# Substantial Copper (Cu^2+^) Uptake by Metakaolin-Based Geopolymer and Its Resistance to Acid Leaching and Ion Exchange

**DOI:** 10.3390/polym15081971

**Published:** 2023-04-21

**Authors:** Nenad Grba, Cyrill Grengg, Mirjana Petronijević, Martin Dietzel, Andre Baldermann

**Affiliations:** 1Department of Chemistry, Biochemistry and Environmental Protection, Faculty of Sciences, University of Novi Sad, Trg Dositeja Obradovica 3, 21000 Novi Sad, Serbia; nenad.grba@dh.uns.ac.rs; 2Institute of Applied Geosciences and NAWI Graz Geocenter, Graz University of Technology, Rechbauerstraße 12, 8010 Graz, Austria; cyrill.grengg@tugraz.at (C.G.); martin.dietzel@tugraz.at (M.D.); 3Faculty of Technology Novi Sad, University of Novi Sad, Bulevar cara Lazara 1, 21000 Novi Sad, Serbia; mirjana.petronijevic@uns.ac.rs

**Keywords:** metakaolin-based geopolymer, heavy metals, environmental protection, copper, water treatment, green technology

## Abstract

Geopolymers are inorganic, chemically resistant aluminosilicate-based binding agents, which remove hazardous metal ions from exposed aqueous media. However, the removal efficiency of a given metal ion and the potential ion remobilization have to be assessed for individual geopolymers. Therefore, copper ions (Cu^2+^) were removed by a granulated, metakaolin-based geopolymer (GP) in water matrices. Subsequent ion exchange and leaching tests were used to determine the mineralogical and chemical properties as well as the resistance of the Cu^2+^-bearing GPs to corrosive aquatic environments. Experimental results indicate the pH of the reacted solutions to have a significant impact on the Cu^2+^ uptake systematics: the removal efficiency ranged from 34–91% at pH 4.1–5.7 up to ~100% at pH 11.1–12.4. This is equivalent to Cu^2+^ uptake capacities of up to 193 mg/g and 560 mg/g in acidic versus alkaline media. The uptake mechanism was governed by Cu^2+^-substitution for alkalis in exchangeable GP sites and by co-precipitation of gerhardtite (Cu_2_(NO_3_)(OH)_3_) or tenorite (CuO) and spertiniite (Cu(OH)_2_). All Cu-GPs showed excellent resistance to ion exchange (Cu^2+^ release: 0–2.4%) and acid leaching (Cu^2+^ release: 0.2–0.7%), suggesting that tailored GPs have a high potential to immobilize Cu^2+^ ions from aquatic media.

## 1. Introduction

Heavy metals, such as lead (Pb), copper (Cu), cadmium (Cd) and mercury (Hg), are hazardous environmental pollutants in Earth’s surface settings, considering their negative impact on human health and ecosystems [1,2,3,4,5]. Most heavy metals are persistent, non-biodegradable and tend to cumulate in, e.g., soils, sediments, surface water, groundwater and living organisms, causing disorders and diseases [6,7,8,9]. Although being an essential microelement and dietary element to plants, animals and humans, Cu compounds are often used as antimicrobial agents to prevent microbial growth on materials’ surfaces, e.g., as coatings in roofing, sheathing and plumbing. However, at higher concentrations, i.e., >5–10 mg Cu^2+^/day in food [10] or >1.3 mg/L Cu^2+^ in drinking water [11], Cu^2+^ possesses strong biostatic properties and a high toxicity, leading to Cu^2+^ poisoning in severe cases [12]. Consequently, in several studies, Cu is proposed to be added to the European Watch List of emerging substances due both to its toxicity potential and increasing abundance in aqueous media, soils and sediments [13,14]. Thus, there is a high demand for the development of efficient, low-cost and eco-friendly (‘green’) cleaning agents in order to minimize the harmful effects of Cu^2+^ in aquatic media [15].

A large number of techniques can be used for the removal of Cu^2+^ from aqueous media, such as adsorption [16], chemical precipitation [17], bioremediation [18], electrokinetic processes [19] and ion exchange [20], etc. These water treatment technologies greatly differ in, e.g., efficiency, operational costs, material and resource uses, waste production and sustainability aspects. So far, only a few attempts have been made to sequester dissolved Cu^2+^ directly in secondary minerals that possess a high affinity towards Cu^2+^ uptake, such as Cu-chelates [21], Fe-(hydr)oxides [22], Cu-(oxy)hydrate chlorides and Cu-carbonates [23,24], Cu-sulfate hydrates [25], calcium-aluminum-silicate-hydrates (C-A-S-H) [26] and geopolymers [27,28]. In particular, the latter bear a high potential for heavy metal immobilization [29,30,31,32] through chemical immobilization.

Geopolymers, when formed using low-calcium (low-Ca) aluminosilicates, are inorganic binders that form three-dimensional, non-crystalline aluminosilicate structures with exchangeable sodium (Na^+^) or potassium (K^+^) ions balancing the negative charges arising from the tetrahedral substitutions of aluminum (Al^3+^) for silicon (Si^4+^) [33]. They are typically formed by reacting metakaolin and/or other inorganic materials having pozzolanic properties, such as fly ash and calcinated clay, with water glass and alkali hydroxide solutions [15]. Their interconnected, micro- to nano-porous structure is advantageous for (ad)sorption processes and metal ion incorporation [31]. Further, geopolymers possess an excellent resistance to temperature [34,35] and chemical corrosion [36,37], rending their application in environmental remediation possible. For example, geopolymers have been successfully utilized for Cu^2+^ [31], Pb^2+^, zinc (Zn^2+^), Cd^2+^ [38,39], manganese (Mn^2+^) and cobalt (Co^2+^) [29] removal from aquatic media, both by chemical precipitation and structural incorporation.

In this study, the immobilization of Cu^2+^ ions by granulated, metakaolin-based geopolymer, as well as their mineralogical and chemical properties and subsequent resistance to ion exchange and leaching, were investigated using a set of equilibrium-approaching batch experiments run under variable pH conditions at ambient temperature. The efficiency and the main reaction mechanisms underlying Cu^2+^ uptake are assessed using a multi-methodological approach. Implications for water treatment and environmental remediation are discussed. Our results provide novel insights into the Cu^2+^ immobilization mechanisms in acidic vs. alkaline aqueous media, which may open the door for future applications of tailored geopolymers for water treatment applications.

## 2. Materials, Experimental Procedures and Methods

### 2.1. Materials

A granulated, metakaolin-based geopolymer (GP) was produced by reacting a liquid alkaline activator (water glass; solid content: 45%; SiO_2_/K_2_O molar ratio = 1.5; pH 13.5, Wöllner Austria GmbH, Gratwein, Austria) with metakaolin powder (SiO_2_/Al_2_O_3_ mass ratio = 1.3 ± 0.1; dry density: 2.6 g/cm^3^, NEWCHEM GmbH, Vienna, Austria) at a liquid/solid mass ratio of 1.7. The fresh paste was homogenized quickly at 250 rpm for 60 s and then slowly at 100 rpm for 30 s using a mixer (N50, HOBART GmbH, Offenburg, Germany). The paste was transferred into standard cubes (size: 10 × 10 × 10 cm) and hardened for 4 days at 20 °C and 50–60% relative humidity. Subsequently, GP was crushed using a jaw crusher. The grain size fractions < 0.125 mm and 0.125–1 mm were separated by sieving, dried at 40 °C and merged in equivalent weights to ensure a homogenous GP product.

The Cu^2+^ solutions were prepared from the dissolution of adequate amounts of copper(II) nitrate trihydrate (Cu(NO_3_)_2_∙3H_2_O, p.a., Merck KGaA, Darmstadt, Germany) in ultrapure water (Milli-Q Plus UV, 18.2 MΩ at 25 °C).

All experiments described below were run in duplicate and the collected liquid samples were analyzed in triplicate. Data reproducibility and accuracy were verified by standard mathematical procedures, yielding < 3% deviation among all experimental sets. Thus, in the following, only the average values are reported.

### 2.2. Experimental Procedures

#### 2.2.1. Cu Immobilization Experiments

The GP was reacted with Cu^2+^-bearing solutions at a fluid/solid ratio of 20:1 in 250 mL polyethylene (PE) reactors at 23 ± 2 °C. The Cu/GP mass ratio was set to 1:2, 1:4, 1:8, 1:13 and 1:25, so that these experiments are labeled as Cu:GP-1:2 to Cu:GP-1:25 (henceforth called Cu:GP series). No pH drift corrections were made. All suspensions were stirred at 200 rpm for 24 h. Fluid samples were taken regularly to follow the temporal uptake of Cu^2+^ by GP. Reference solutions containing Cu^2+^ in the above concentration range (but without GP) were prepared to demonstrate that other mechanisms causing Cu^2+^ removal from solution are negligible. Further, a reference experiment with GP but without dissolved Cu^2+^ was made to determine the pH development of pure GP in ultrapure water, which equilibrated at pH ~11.8. Following up, the effect of pH treatment on Cu^2+^ immobilization by GP was studied. The experimental set-up was identical to those described above, except for the adjustment of the solution pH to ~11–12 by adding droplets of 1 M sodium hydroxide solution (NaOH, p.a., Roth). This treatment caused less than ~2% volume change to the solutions, so that no corrections to the dissolved Cu^2+^ concentrations were made. These experiments are labeled as Cu:GP_pH_-1:2 to Cu:GP_pH_-1:25 (henceforth called Cu:GP_pH_ series).

All experiments were terminated after 24 h. The solids were separated by 0.45 µm cellulose acetate membrane filters (Sartorius, Göttingen, Germany) using a suction filtration unit, rinsed with ultrapure water to remove electrolytes and dried at 40 °C in preparation for solid-phase analyses. Aliquots were stored under an argon (Ar) atmosphere for subsequent use in ion exchange and leaching experiments. All liquids were acidified using suprapure nitric acid (HNO_3_, ROTIPURANR, Roth, Karlsruhe, Germany) in preparation for elemental analyses.

#### 2.2.2. Ion Exchange Experiments

Ion exchange experiments were carried out to determine the portion of exchangeable Cu^2+^ in GP. Therefore, 2 g Cu-GPs obtained from the Cu:GP and Cu:GP_pH_ series were treated with 200 mL of a 100 mM sodium chloride (NaCl) solution (fluid/solid ratio = 100:1) for 24 h in 250 mL PE reactors at 23 ± 2 °C. The experiments were stirred continuously at 250 rpm. Fluid samples were taken regularly to determine the Cu^2+^ exchange dynamics of GP until chemical steady-state conditions were reached.

#### 2.2.3. Leaching Experiments

Leaching tests were conducted to quantify the fraction of weakly bonded Cu^2+^ in GP using a modified Toxicity Characteristic Leaching Procedure (TCLP) [40]. For this purpose, 2 g Cu-GP obtained from the Cu:GP and Cu:GP_pH_ series were reacted with 200 mL ultrapure water (fluid/solid ratio = 100:1) set to pH ~4.5 with the use of 0.5 M hydrogen chloride solution (HCl, p.a., Merck KGaA) in 250 mL PE reactors at 23 ± 2 °C. Experiments were stirred at 250 rpm and lasted for 72 h to ensure chemical steady state. Fluid samples were taken regularly to determine the Cu^2+^ release from GP upon acid leaching.

### 2.3. Analytical Methods

#### 2.3.1. Fluid-Phase Characterization

Temperature, electric conductivity (EC) and pH were determined with WTW LF/pH 330 m multi-meters connected to TetraCon 325 and SenTix41 probes. The pH electrodes were calibrated against NIST buffer standard solutions at pH 4.01, 7.00 and 10.01 at 25 °C at an analytical precision of ±0.05 pH units [41].

Copper, K^+^ and Na^+^ concentration analyses were made on a PerkinElmer Optima (PerkinElmer, Waltham, MA, USA) 8300 DV inductively coupled plasma optical emission spectrometer (ICP-OES) on acidified samples. NIST 1640a, in-house and SPS-SW2 Batch 130 standards were analyzed within repeated sample sequences, yielding an analytical error of ±3% and a detection limit of <0.01 mg/L for each element of interest [42]. From these chemical data, the removal efficiency (%removal) and the amount of Cu^2+^ immobilized by GP (q_e_ in mg/g) were calculated following Equations (1) and (2):%removal = ((c_0_−c_e_)·c_0_^−1^)·100(1)
q_e_ = ((c_0_−c_e_)·m^−1^)·V(2)
where c_0_ and c_e_ denote the initial and equilibrium Cu^2+^ concentrations in solution (in mg/L), m is the dry mass of GP (in g) and V is the volume of the solution (in L).

#### 2.3.2. Solid-Phase Characterization

The mineralogy of GP and all Cu-GP samples was determined by X-ray diffraction (XRD) using a PANalytical X’Pert Pro operated at 40 kV and 40 mA (Co-Kα) and outfitted with a Scientific X’Celerator detector. The powdered specimens were prepared using the top loading technique and examined in the 4–85° 2θ range using a step size of 0.008° 2θ and 40 s count time per step. Further, the crystallinity of the Cu-bearing co-precipitates (see below) was determined by means of the full width at half-maximum (FWHM in ° 2θ) values obtained from the respective peak of each phase with the highest diffracted intensity. The PANanalytical X’Pert Highscore Plus software and a pdf-4 database were used for the interpretation of the XRD patterns [43].

Fourier transform infrared (FTIR) spectroscopy of GP was carried out in Attenuated Total Reflectance (ATR) mode using a PerkinElmer Frontier FTIR. Mid-infrared (MIR) spectra were collected in the 650–4000 cm^−1^ range at a resolution of 2 cm^−1^ [44]. Data processing was made via the Spekwin32 software (version 1.71.6.1).

The major elemental composition of GP and of some Cu-GPs was analyzed with a PANalytical PW2404 wavelength dispersive X-ray fluorescence (XRF) spectrometer. Glass tablets were prepared in a PANalytical Perl’X 3 bead preparation system by the fusion of 0.5 g material (pre-dried at 110 °C) with 6.0 g lithium tetraborate (Li_2_B_4_O_7_, Malvern Panalytical, Malvern, UK) at 1200 °C. The loss on ignition (LOI) was determined by gravimetric analysis of material residues glowed at 1050 °C for 1 h. The analytical error is <0.5 wt.% for the major elements, as determined by replicate analyses of USGS standards [43]. Comparison of Cu contents of Cu-GPs determined by ICP-OES and XRF analyses yielded a positive linear correlation with a slope of 0.98, an intercept of 0.1 and a R^2^ value of 0.95 (*n* = 9), so that only the ICP-OES data are presented in the following.

Secondary electron (SE) images were acquired on a ZEISS DSM 982 Gemini scanning electron microscope (SEM) (ZEISS, Jena, Germany) operated at an accelerating voltage of 3–5 kV for microstructural characterization of the GP and Cu-GP samples. Representative material was prepared on SEM stubs, fixed with double-sided carbon (C) tape and subsequently C-coated to reduce charging [45]. Energy-dispersive X-ray spectroscopy (SEM-EDX) data were acquired on selected single spots using 15 kV accelerating voltage and subsequently used for mineral identification and quantification of the Cu content of the GP matrices. Chemical data (*n* = 5/sample; average values and two standard deviations (2 SD) are reported) were obtained from element k-factors determined on mineral standards at an analytical precision of <1 at.% for K, Na, Al, Si and Cu analyses [26].

The specific surface area (SSA) of the GP was measured before and after the batch experiments by the multi-point adsorption Brunauer, Emmett and Teller (BET) method using a Micromeritics FlowSorb II2300 (Micromeritics, Norcross, GA, USA) and a He(69.8)-N_2_(30.2) mixture as the carrier gas (analytical error: ±10%).

## 3. Results and Discussion

### 3.1. Geopolymer Characterization

The granulated GP (Figure 1a) had a Si:Al molar ratio (SAR) of 1.79 and comprised mainly of 45.5 wt.% SiO_2_, 22.4 wt.% Al_2_O_3_, 15.6 wt.% K_2_O, 13.6 wt.% LOI, 1.1 wt.% Fe_2_O_3_, 1.0 wt.% Na_2_O and 0.8 wt.% TiO_2_, which is typical for conventional geopolymers made of metakaolin [46]. The CuO content of GP was determined to be 55 ppm only. The mineralogy (Figure 1b) was dominated by an amorphous phase (95 wt.%) due to the non-crystalline aluminosilicate structure of conventional geopolymers [33], in addition to quartz (4 wt.%) and anatase (1 wt.%) that are inherited from the metakaolin raw material. The FTIR spectrum of GP (Figure 1c) revealed a broad absorption at 3385 cm^−1^ and a more intense IR band at 1641 cm^−1^, which are attributed to OH stretching vibrations and H-O-H bending vibrations of interlayer adsorbed water and structurally bound water molecules in geopolymers [47]. The weak adsorption at 1381 cm^−1^ indicates the presence of minor amounts of potassium carbonate (K_2_CO_3_) in GP, which probably formed by the interaction of potassium hydroxide (KOH), water and atmospheric carbon dioxide (CO_2_) during geopolymer preparation [48]. Additional IR bands at 972 cm^−1^, 881 cm^−1^ and 684 cm^−1^, respectively, can be assigned to asymmetric stretching vibrations of Si-O-Al and stretching vibrations of Si-O in Si-OH groups [48,49].

The GP grains had an average size between 100–200 µm, an external SSA value of 2.18 m^2^/g and appeared as particle agglomerates (Figure 1d), which is typical for metakaolin-based geopolymer granulates [50]. The GP microstructure was largely homogeneous, as indicated by the uniform distribution of a porous aluminosilicate matrix (Figure 1e), which barely contained unreacted quartz grains and anatase crystals. SEM-EDX analyses confirmed the potassium-aluminum-silicate-hydrate (K-A-S-H) composition of the geopolymer matrix, as evident from SiO_2_/Al_2_O_3_ ratios of ~2.0 and Al_2_O_3_/K_2_O ratios of ~1.4. According to Kim and Lee [51], geopolymers with high Si/Al ratios have a more uniform, finer and more smoothly connected microstructure than those with low Si/Al ratios, rendering the GP product studied here suitable for water treatment applications.

### 3.2. Cu^2+^ Immobilization by Geopolymers

The temporal evolution of the dissolved Cu^2+^ concentrations for the experiments of the Cu:GP (without pH adjustment) and Cu:GP_pH_ (with alkaline treatment) series is illustrated in Figure 2. A fast decrease of the Cu^2+^ concentration was seen in all experiments within <60 min, which was followed by a slowly decreasing trend thereafter (Figure 2a,b) until chemical equilibrium conditions were attained after ~15 h, judged from the establishment of constant EC values until the experiments were terminated after 24 h. The Cu^2+^ removal was generally higher in the Cu:GP_pH_ series yielding final Cu^2+^ concentrations below the recommended value for Cu^2+^ in drinking water (<1.3 mg/L) [11]. Thus, the alkaline treatment resulted in high removal efficiencies for Cu^2+^ in the case of the Cu:GP_pH_ series (about 100%), whereas the Cu:GP series exhibited moderate to high Cu^2+^ removal efficiencies, ranging from 34 to 91% (Figure 2c,d). The fate of Cu^2+^ in the experiments of the Cu:GP series depended on the initial Cu:GP ratio used, i.e., the Cu^2+^ stock solutions had pH values between 3 and 5 before the reaction with GP and thus on the final solution pH, which varied between 4.1 and 5.7. This suggests that the reactivity of Cu^2+^ was largely pH dependent. This observation supports the conclusions drawn by Potysz et al. [52], who showed that strongly acidic conditions (pH 2) favor leaching and dissolution of Cu-bearing materials, such as Cu slags, whereas close to neutral (pH 7–8) and alkaline (pH 12–13) conditions enhance the stability of Cu-bearing minerals.

The amount of Cu^2+^ immobilized in the Cu:GP and Cu:GP_pH_ series ranged from about 40 to 193 mg/g and 44 to 560 mg/g, respectively (Figure 2e,f). Higher Cu^2+^ concentrations did not result in elevated Cu^2+^ removal capacities, which indicates that the highest q_e_ values present approximately the maximum removal capacity (‘q_m_’) of the GP mixture considered in this study. However, we note that co-precipitation of Cu minerals occurred in all experimental series, so that we cannot apply ‘classical’ adsorption isotherm models and pseudo-first-order or pseudo-second-order kinetic models to further fit our experimental data. The ‘q_m_’ value obtained for experiment Cu:GP-1:2 is 27% higher than Cu^2+^ removal by an amorphous geopolymer synthesized from fly ash (152 mg/g at 45 °C) [27] and 79% higher than the amount of Cu^2+^ removed from solution by an organically modified, metakaolin-based mesoporous geopolymer (108 mg/g at 30 °C) [53]. Moreover, Cu^2+^ uptake by GPs with(out) alkaline conditioning was much higher than prior reported for other geopolymeric substances [54]. Experiment Cu:GP_pH_-1:2 yielded a q_e_ value for Cu^2+^ uptake in the same order of magnitude than previously reported for Pb^2+^ uptake by porous geopolymer-based microspheres (629 mg/g at 25 °C) [55]. This comparison underlines the efficiency of GP for Cu^2+^ uptake from solution.

### 3.3. Mineralogy and Microstructure of Cu^2+^-Bearing Geopolymers

XRD and SEM analyses of the precipitates from the Cu:GP and Cu:GP_pH_ series, obtained after the Cu^2+^ immobilization experiments, identified Cu^2+^-substituted GP, in addition to unreacted quartz and anatase and three types of newly formed Cu^2+^-bearing minerals (Figure 3). All Cu-GP materials collected after ion exchange and leaching tests were also studied by XRD, but they did not reveal mineralogical changes and are therefore not shown. The external SSA values of all reacted GPs varied between 1.98 and 2.35 m^2^/g, which is within the analytical uncertainty of the BET measurements.

The copper nitrate hydroxide gerhardtite (Cu_2_(NO_3_)(OH)_3_) precipitated in varying proportions under the acidic conditions (pH 4.1 to 5.7) of the Cu:GP series (Figure 3a), confirming that this mineral is the most stable polymorph at ambient temperature of the copper hydroxyl nitrate family, which include gerhardtite, rouaite and likasite [56]. The formation of well-crystallized gerhardtite (FWHM: 0.13–0.18° 2θ) particles with an average size of 0.5 to 1.0 µm and a plate-like morphology on the GP grains and within the pore space (Figure 3b) can be explained by its lower solubility compared to the copper(II) nitrate trihydrate salt, used for the preparation of the Cu^2+^ solutions and a related high supersaturation of gerhardtite with respect to the acidic solutions of the Cu:GP series [57]. However, a considerable amount of Cu^2+^ was also incorporated in the GP matrix, as evident from the CuO/Al_2_O_3_ ratio of 0.3 ± 0.1 in experiment Cu:GP-1:25, where no gerhardtite formed. All other GP matrices had a CuO/Al_2_O_3_ ratio of 0.2 ± 0.1. The amount of Cu^2+^, which was bound to gerhardtite, increased at elevated initial Cu:GP ratio (Figure 3a). Thus, the low solubility of gerhardtite most likely reduced the uptake capacity of Cu^2+^ by GP.

The copper(II) (hydr)oxide minerals spertiniite (Cu(OH)_2_) and tenorite (CuO) formed under the highly alkaline conditions (pH 11.1 to 12.4) of the Cu:GP_pH_ series (Figure 3c). This finding is supported by experimental evidence of copper(II) (hydr)oxides formation controlling the Cu^2+^ concentrations in aqueous solutions containing hydroxide (OH^-^) and nitrate (NO_3_^−^) or sulphate (SO_4_^2−^) as sole anions [58,59]. Geochemical equilibrium modeling further indicated that CuO and Cu(OH)_2_ phases can limit the metal ion mobility in cement and geopolymer matrices under alkaline conditions [60]. The formation of plate-like sper-tiniite (~0.1–0.2 µm in largest dimension) and of blocky tenorite (~0.2–0.5 µm in size) was controlled mainly by the pH treatment applied to this experimental set (Figure 3d). Thus, a large fraction of Cu^2+^ added to the Cu:GP_pH_ series was immobilized by barely soluble copper(II) (hydr)oxides. However, a portion of Cu^2+^ was chemically bound to GP, as evidenced by the CuO/Al_2_O_3_ ratio of 0.20 ± 0.1 in experiment Cu:GP_pH_-1:25, where the amount of semi-crystallized tenorite (FWHM: 0.77–0.89° 2θ) and poorly crystallized spertiniite (FWHM: 1.2–1.6° 2θ) formed is negligible (Figure 3c). All other GPs had a CuO/Al_2_O_3_ ratio of 0.15 ± 0.5, signifying profound Cu^2+^ incorporation in the GP structure.

### 3.4. Chemical Resistance of Cu^2+^-Bearing Geopolymer

#### 3.4.1. Ion Exchange

The ion exchange behavior of the Cu-GP materials is illustrated in Figure 4. We note that a small amount of K^+^ (<0.2 mmol/L) was liberated to the experimental solutions from all Cu-GPs. This could represent either a mobile fraction of K^+^ that occupied surface-accessible sites in Cu-GPs or dissolution of K_2_CO_3_ impurities present in the raw GP [48]. However, no changes in BET-SSA of all ion-exchanged GPs were recognized within the analytical uncertainty of the BET measurements.

As for the Cu:GP series, a minor proportion of Cu^2+^ (0.03–0.74 mmol/L) was exchanged for Na^+^ (0.17–1.39 mmol/L) upon reaction with a 100 mM NaCl solution for 24 h (Figure 4a). This indicates that some Cu^2+^ ions occupied easily exchangeable (surface) sites in the Cu-GPs prepared under acidic conditions (Figure 2c). The Cu^2+^/Na^+^ molar ratio was close to 0.5, which is needed to account for a charge compensation in the Na^+^-substituted Cu-GPs (Figure 4a). Such stoichiometric exchange of Cu^2+^ for Na^+^ suggests that the dissolution of gerhardtite was negligible and that a fraction of the initial Cu^2+^ concentration added to the Cu:GP series was incorporated in the GP structure [57], corroborating the SEM-EDX results. The proportion of Cu^2+^ dissolved from Cu-GPs and subsequently returned back to the solutions varied only between 0.4 and 2.4% (Figure 4b), which in turn demonstrates the low reactivity of the Cu-GPs during ion exchange reactions. Accordingly, SE images collected on ion-exchanged Cu-GPs show platy, well-crystallized gerhardtite (FWHM: 0.14–0.19° 2θ) particles that co-occur with the Na^+^-substituted Cu-GPs having a CuO/Al_2_O_3_ ratio of 0.3 ± 0.2 (Figure 4c).

As for the Cu:GP_pH_ series, a small fraction of K^+^ was exchanged for Na^+^ (<0.2 mmol/L) to balance the charge in the Cu-GPs, similar to the materials of the Cu:GP series. In contrast, the Cu^2+^/Na^+^ molar ratio was determined to be only ~0.1, so that no correlation between exchanged Na^+^ and Cu^2+^ ions was recognized (Figure 4d), which underlines the low reactivity of Cu^2+^ bound to geopolymers under highly alkaline conditions [52,60]. Consequently, the fraction of Cu^2+^ mobilized from the materials of the Cu:Gp_pH_ series upon Na^+^ exchange was very low (<0.04%; Figure 4e), which again demonstrates the immobility of Cu^2+^ in the Cu:Gp_pH_ series. The SE images collected show no microstructural modifications compared to the untreated Cu-GPs (Figure 4f) and the K-A-S-H matrices had a CuO/Al_2_O_3_ ratio of 0.2 ± 0.1.

#### 3.4.2. Acid Leaching

The resistance of the Cu-GPs to acid leaching (pH 4.5) is shown in Figure 5. The Al^3+^, Fe^3+^, Na^+^, Si(OH)_4_ and Cu^2+^ concentrations remained always ≤100 mg/L and ≤8 mg/L in the leachate solutions of the Cu:GP and Cu:GP_pH_ series, which proves the high acid resistance of all the Cu-GPs. The K^+^ concentration ranged between 30 and 145 mg/L, equivalent to a dissolution quota of ~1–2 wt.%, if all K^+^ is assigned to K_2_CO_3_ impurities present in the GP (Figure 1b). This K_2_CO_3_ dissolution caused a slight increase in solution pH, from 5.1 to 5.6, in all experiments. Nevertheless, the BET-SSA and the crystallinity degree (FWHM_gerhardtite_: 0.13–0.18° 2θ; FWHM_tenorite_: 0.75–0.91° 2θ; FWHM_spertiniite_: 1.1–1.5° 2θ) of all leached GPs remained unchanged within the analytical uncertainty of the BET and XRD measurements.

The fraction of Cu^2+^ released back to the leachate solutions varied between 0.24 and 0.68% for the materials of the Cu:GP series (Figure 5a), which indicates that the Cu-GPs, including the co-precipitated gerhardtite, have a high resistance to leaching. This finding is consistent with the experimentally determined solubility minimum of gerhardtite under slightly acidic to near-neutral conditions (pH ~5.5 to 7.5) and a low solubility product (K_gerhardtite_ = ~10^−16^; 25 °C) of this barely soluble mineral [61]. Accordingly, SE images taken from the leached Cu-GPs show only minor microstructural changes, such as a small increase of fine pores that were formerly filled by gerhardtite (Figure 5b). SEM-EDX analyses revealed no chemical alteration of the K-A-S-H matrices (i.e., CuO/Al_2_O_3_ ratio = 0.3 ± 0.2) upon leaching.

The amount of Cu^2+^ measured in the leachate solutions ranged from 0.23 to 0.65% for the materials of the Cu:GP_pH_ series and increased slightly with increasing initial Cu:GP_pH_ ratio (Figure 5c), which suggests that a partial dissolution of CuO and Cu(OH)_2_ phases accounted for the Cu^2+^ release. This assertion is supported by the lower stability of these minerals under acidic conditions and their higher solubility in pure water (K_tenorite_ = 10^−8.6^; K_spertiniite_ = 10^−11.3^; 25 °C) compared to gerhardtite [58,62,63,64]. No visible microstructural modifications were recognized for the leached Cu-GPs of the Cu:GP_pH_ series (Figure 5d). The K-A-S-H matrices had a CuO/Al_2_O_3_ ratio of 0.2 ± 0.1.

### 3.5. Implications for Water Treatment and Environmental Remediation

Ion exchange and leaching reactions had only a minimal impact on the mineralogical, chemical and microstructural composition of the Cu-GPs obtained from the immobilization experiments with(out) alkaline treatment (Figure 4 and Figure 5). This indicates that the vast majority of the Cu^2+^ ions was bound to ‘unreactive’, barely soluble mineral phases, such as gerhardtite under acidic conditions versus tenorite and spertiniite under alkaline conditions (Figure 3a,c), in addition to the Cu^2+^-substituted in the GP matrix. The low reactivity of all Cu^2+^-containing phases under the herein tested conditions suggests that a high immobilization degree of Cu^2+^ was achieved, i.e., only ≤2.4% of the Cu^2+^ bound within the Cu-GPs was remobilized (Figure 4b,e and Figure 5a,c). This highlights the potential of granulated GP for water treatment applications, especially in acidic aqueous media where sorbent materials must withstand corrosive conditions [15,16,20]. Moreover, GP can be easily removed by filtration or sedimentation from treated wastewater, ensuring economic use of this novel sorbent material (Figure 6a–c).

Baldermann et al. [65] argue that metal ion (Me^2+^)-bearing C-A-S-H, a common constituent of Ordinary Portland cement, will rapidly decompose even under mildly acidic conditions, thereby providing Ca^2+^ and Me^2+^ ions to the aqueous phase and leaving least soluble solid residues, such as amorphous silica and poorly crystalline aluminosilicates, at the leaching front. This dissolution reaction could thus release harmful Cu^2+^ ions to the aquatic environments if no other immobilization mechanisms are active. Alike, the leaching behavior of geopolymers is known to increase with acidity and especially at pH < 4.5, the structural integrity of the K-A-S-H network becomes increasingly weakened [52]. Nevertheless, our results suggest a low leachability of Cu^2+^ from the Cu-GPs under mildly acidic conditions (Figure 5), which indicates that chemical bonds are responsible for the binding of Cu^2+^ to the molecular structure of the GPs [66]. However, their long-term resistance to corrosion in highly acidic environments (pH < 4) has to be demonstrated, but this is beyond the scope of this study.

The type of binding of Cu^2+^ ions in the three-dimensional molecular network of geopolymers is key for predicting their stability in corrosive systems. The behavior of Cu^2+^-containing low-Ca geopolymers is often viewed analogously to zeolite minerals [67], because of their apparent crystal-chemical similarities. Zeolites are scaffold silicates, consisting of cross-linked (Si,Al)O_4_ tetrahedrons with Na^+^ or K^+^ ions occupying exchangeable sites to keep electrostatic neutrality. Natural and synthetic zeolites can immobilize various pollutants (e.g., radioactive components and heavy metal ions) through ion exchange [15]. Our results indicate that the Cu-GPs had a high resistance against Na^+^ exchange (Figure 4), suggesting that the exchange processes with other Me ions are also limited. On the other hand, the porous geopolymer network may allow the sorption and/or incorporation of potentially other hazardous Me cations from solution given that these dissolved components have an identical ionic charge and a Me ionic radius similar to Cu^2+^, such as Co^2+^, Mn^2+^, nickel (Ni^2+^), palladium (Pd^2+^) and Zn^2+^ [67]. However, the immobilization potential of such Me ions by GP has to be evaluated in future studies, with particular focus on near-neutral and acidic aqueous media where Me ion immobilization by GP is crucial. Further studies should also consider the impact of temperature, (ad)sorption kinetics, presence of competing ions in solution and renewability/stability of GP under certain conditions in order to make progress in yet untested geopolymer applications in diverse water treatment scenarios at full-industrial scale.

The use of Cu^2+^-substituted geopolymers is increasingly considered as a novel, efficient technology to enhance the antimicrobial properties of building and construction materials [68]. These chemically resistant aluminosilicate structures may even withstand the aggressive conditions frequently met in wastewater systems [33]. They exhibit good antimicrobial effects on Cu^2+^-treated geopolymer surfaces and a sufficient in vitro stability in the long term [69,70]. Hashimoto et al. [70] have demonstrated that the antimicrobial activity of metakaolin-based geopolymers against fungi hyphae can be increased by the incorporation of Cu^2+^ into the geopolymer structure. Our results proved that this chemical substitution can reach CuO/Al_2_O_3_ ratios of 0.3 ± 0.1 in the GP matrix under the experimental conditions used in this study. The presence of additional CuO/Cu(OH)_2_ phases on geopolymer surfaces and within fine pores could further increase their bactericidal activity (Figure 3), as indicated by a biological study using S. aureus and E. coli bacterial strands treated with Cu(OH)_2_ nanoparticles and mixed Cu_1−x_Mg_x_(OH)_2_ nanorods, i.e., a bacterial cell reduction of >99.99% was observed after 180 min at room temperature [64]. Thus, GP has a high potential to be used for advanced and large-scale water treatment and purification studies after successful testing and calibration against latest wastewater purification technologies [71,72,73,74,75].

## 4. Conclusions and Perspectives

A granulated geopolymer (GP) was prepared from an alkali-activated metakaolin powder and subsequently exposed to copper (Cu^2+^)-containing solutions to determine the immobilization potential of the GP with(out) pH adjustment. The Cu^2+^ removal efficiency depended mainly on the initial Cu:GP ratio and on the pH and reached >99.9% under alkaline conditions and up to 91% under acidic conditions, which is equivalent to Cu^2+^ removal capacities of 193 mg/g and 560 mg/g, respectively. A substantial fraction of Cu^2+^ immobilization was due to chemical incorporation into the GP matrix, but a large amount of Cu^2+^ was also precipitated in the form of gerhardtite (low pH) or tenorite and spertiniite (high pH). The resistance of the Cu^2+^-bearing GPs to leaching and ion exchange was experimentally verified, yielding a small Cu^2+^ release of less than 2.4%. Accordingly, geopolymers bear a high potential to consistently immobilize aqueous Cu^2+^. Further work should (i) make use of the herein described novel Cu-GPs and explore their performance and durability in corrosive settings at field scale and (ii) explore the links between crystal structure and Cu-binding environments in Cu^2+^-substituted geopolymers using e.g., solid-state nuclear magnetic resonance (^29^Si and ^27^Al MAS-NMR) or chemical-state X-ray photoelectron spectroscopic analysis (XPS), both to gain insights into the resistance of Cu-bearing geopolymers to natural and man-made corrosion settings.

## Figures and Tables

**Figure 1 polymers-15-01971-f001:**
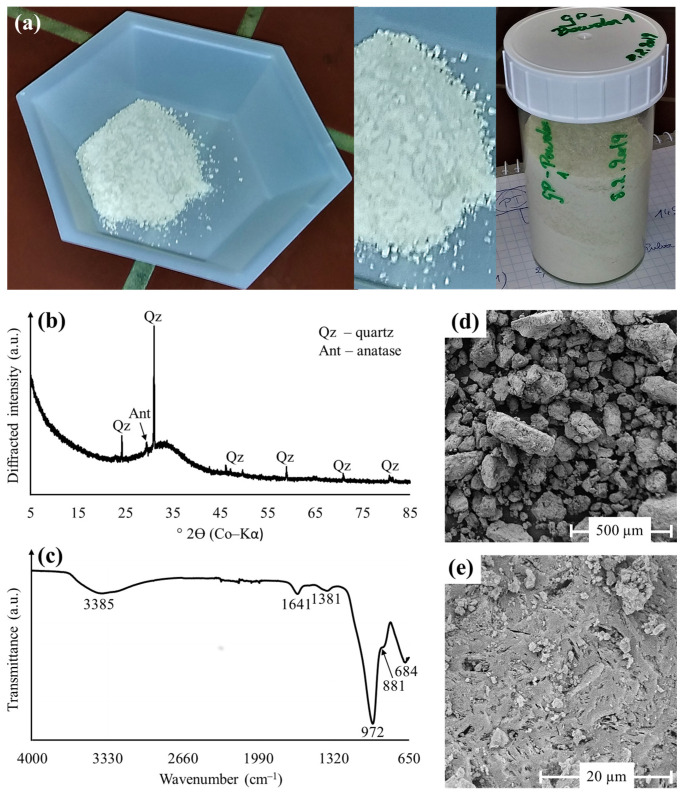
Optical appearance, mineralogical composition and microstructural features of granulated pristine GP used for the Cu^2+^ immobilization studies. (**a**) Photograph; (**b**) XRD pattern; (**c**) FTIR spectrum; (**d**,**e**) SE images of GP.

**Figure 2 polymers-15-01971-f002:**
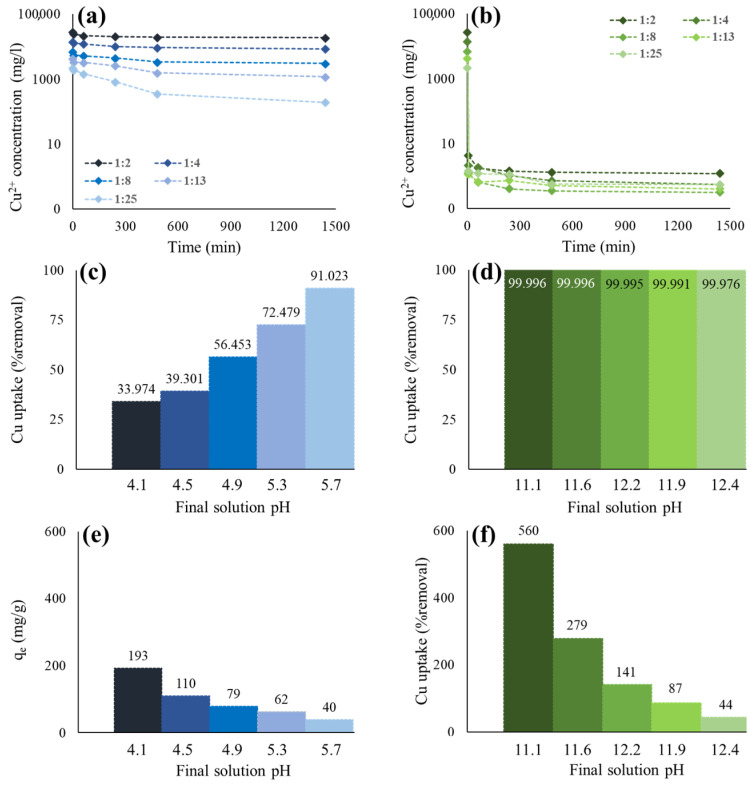
Compilation of results from the Cu^2+^ immobilization experiments. (**a**,**b**) Evolution of the dissolved Cu^2+^ concentration for experiments of the Cu:GP series (left panel) and Cu:GP_pH_ (right panel) series; (**c**,**d**) Fraction of Cu^2+^ immobilized by GP with(out) alkaline treatment; (**e**,**f**) Cu^2+^ removal capacity of the Cu:GP series (left panel) and Cu:GP_pH_ series (right panel).

**Figure 3 polymers-15-01971-f003:**
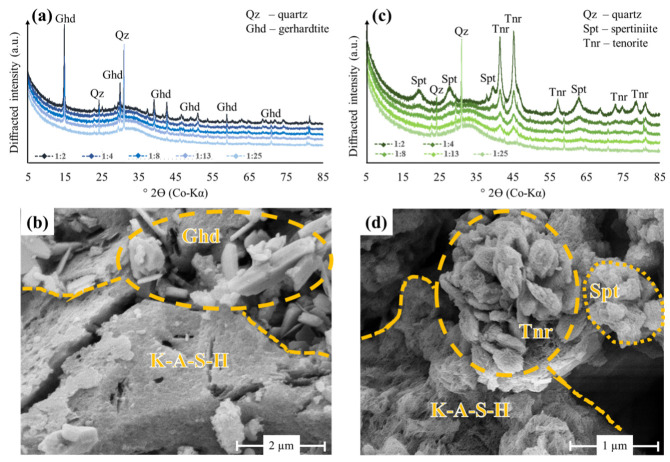
Mineralogy and microstructure of Cu-GPs obtained from the Cu^2+^ immobilization experiments of the Cu:GP series (left panel) and Cu:GP_pH_ series (right panel). (**a**,**c**) XRD patterns of all precipitates; (**b**,**d**) SE images of precipitates from experiments Cu:GP-1:2 and Cu:GP_pH_-1:2, respectively.

**Figure 4 polymers-15-01971-f004:**
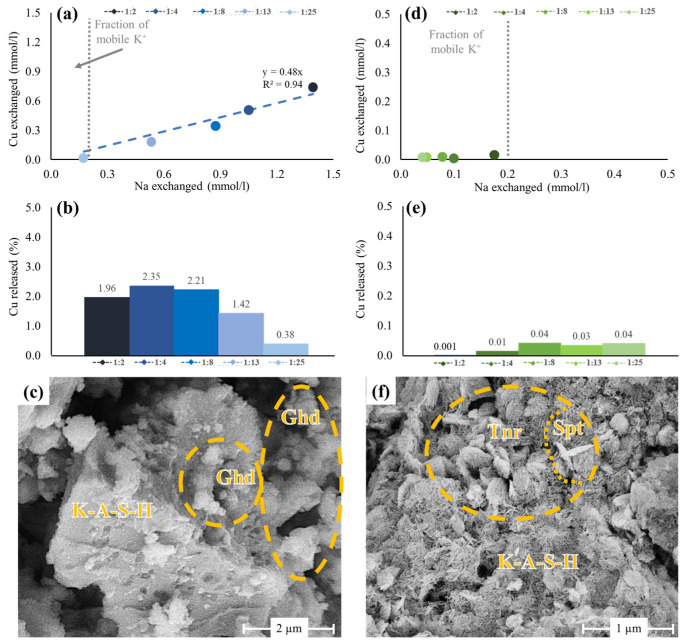
Compilation of results obtained from the ion exchange experiments. (**a**,**d**) Cross-plots of the Cu^2+^ and Na^+^ concentrations showing stoichiometric ion exchange in the Cu:GP series and no ion exchange in the Cu:GP_pH_ series. Note the different scale in (**a**,**d**); (**b**,**e**) Fraction of Cu^2+^ liberated into the solutions upon ion exchange in the Cu:GP and Cu:GP_pH_ series. Note the small amounts of ‘mobile’ K^+^ released back to the solutions and the different scale in (**b**,**e**); (**c**,**f**) The SE images of the Na^+^-exchanged samples Cu:GP-1:2 and Cu:GP_pH_-1:2 reveal no microstructural changes after the ion exchange reaction. Ghd—gerhardtite; Spt—spertiniite; Tnr—tenorite.

**Figure 5 polymers-15-01971-f005:**
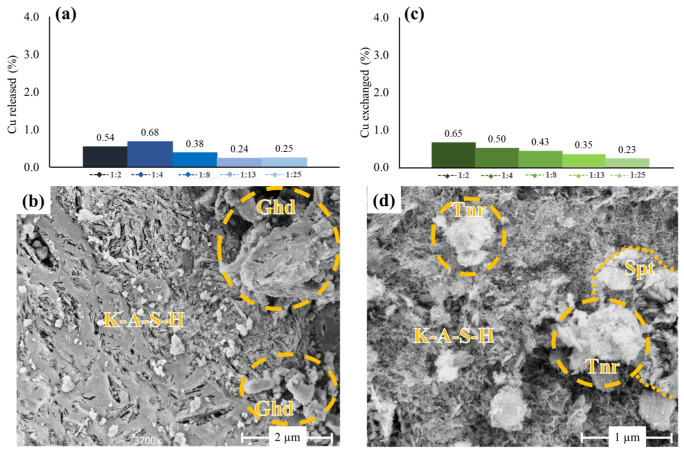
Compilation of results obtained from the leaching experiments. (**a**,**c**) Fraction of Cu^2+^ liberated to the solutions upon acid leaching in the Cu:GP and Cu:GP_pH_ series; (**b**,**d**) SE images of the leached samples Cu:GP-1:2 and Cu:GP_pH_-1:2 revealing no microstructural changes.

**Figure 6 polymers-15-01971-f006:**
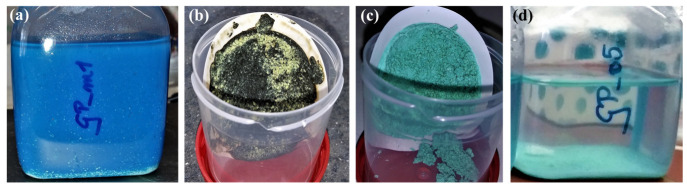
Photo documentation of the efficiency of Cu^2+^ removal from solution by GP. (**a**) Dark blue Cu^2+^-containing solution at the beginning of experiment Cu:GP_pH_-1:2; (**b**,**c**) Precipitates obtained at the end of experiments Cu:GP-1:2 and Cu:GP_pH_-1:2; (**d**) Transparent solution at the end of experiment Cu:GP_pH_-1:2 with a Cu^2+^ concentration below international guideline values.

## Data Availability

All raw data will be made available upon request to the corresponding author.

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
