# Peer review of "Substantial Copper (Cu2+) Uptake by Metakaolin-Based Geopolymer and Its Resistance to Acid Leaching and Ion Exchange"

_polymers, 2023, doi:10.3390/polym15081971_

Round 1
Reviewer 1 Report
The following comments need to be addressed:
1) Lines 597-613 should be deleted.
2) Provide a Graphical Abstract.
3) Provide Highlights.
4) Show images of the GP raw material and of the GP-treated Cu solution.
(additional comments)
1) What does it add to the subject area compared with other published material?A novel granulated geopolymer is reported, which shows good immobilization potential for aqueous Cu in acidic and alkaline matrices.
2) Do you consider the topic original or relevant in the field? Does it address a specific gap in the field?
The content of the manuscript is novel with regards to the presented metal uptake by an inorganic aluminosilicate binder. The text is concise and straight to the point.
3) What specific improvements should the authors consider regarding the methodology? What further controls should be considered?
In my opinion, the experiments have been conducted systematically within the selected scope of the work.
Author Response
Dear Reviewer,
please find our response in the attached pdf file.
Kind regards,
Andre Baldermann

Reviewer 2 Report
The authors utilized a granulated geopolymer (GP) to immobilized copper ions from water under different conditions, e.g., Cu : GP ratio and pH values. Release of Cu from GP was also investigated. The research showed application of a new material for metal resource recovery. Paper can be accepted after minor revisions.
1. The novelty of application of GP to immobilize metals can be emphasized in the Introduction.
2. How is other metals? Can this material be used for other metal resource?
3. It is obviously high pH facilitate the removal of Cu, we may more concern about natural pH or acidic solutions.
4. How are other conditions? E.g., if there are chelation agents in the solution (CN- or humic acid), and competing divalent cations.
Author Response

(The authors gave the same response as above.)

Reviewer 3 Report
Grba et al. investigate the application of metakaolin-based GP as an immobilizing material for heavy metal contaminant – Cu2+. Characteristics of the GP was studied based on FT-IR, XRD, SEM. Before and after its use in Cu2+ was also investigated using XRD and SEM. The resistance of the Cu2+-bearing GP against the leaching and ion exchange was investigated, where only small amounts of Cu2+ (<2.4%) released back to the system.
Major concerns:
1. Though it is a potential study, but the investigation was very limited and I feel uncertain about its publication. The batch adsorption or in this case is the immobilization has not been investigated thoroughly, such as the operating temperature was not investigated for its effect. Characterization should be extended to BET (porosity) and pHpzc to give a comprehensive properties of the material. Moreover, isotherm, kinetics, and thermodynamic studies were not carried out here. The kinetics of the Cu2+ release was also not determined. Altogether, I found these data are minimum for publication.
Minor issues:
1. I think the adsorption tests should be carried out in triplicate, hence presented as mean±standard deviation (SD).
2. Crystallinity and crystallite size should be determined using XRD.
3. In Figure 4&5. Authors compare the SEM images to suggest the non-changing state of the microstructure of the GP. However, SEM images could not provide an objective measurement for this purpose. I suggest additional analysis such as XRD or BET (since we concern about the micropores).
4. Scanning using elemental mapping based on the SEM-EDX could be carried out.
5. In Figure 4(b,e) & 5(a,b). It is not clear about the x-axis.
Author Response

(The authors gave the same response as above.)

Reviewer 4 Report
The authors reported metakaolin-based geopolymer for efficient removal of Cu2+.
The results are interesting and some reasonable explanation is provided. It is acceptable with revision. However, several modifications are required as follows:
1. Pictures need to be clearer and more attractive, to facilitate reading.
2. The SEM images with higher magnification should be provided to clearly real the structure of sample.
3. The BET surface areas of metakaolin-based geopolymer should be measured and analyzed.
4. The advances of wastewater purification technology, including heavy metals removal should be introduced to keep abreast of the latest research trends. e.g.: Chinese Journal of Catalysis, 2022, 43, 2652–2664; Adv. Fiber Mater., 2022, 4, 1620, Chem. Eng. J., 2023, 455, 140943.
5. The advantages and disadvantages of the materials should be discussed and compared with others.
6. There are some publications about the wastewater purification, and the author may cite them in the manuscript to enrich the introduction, for example: Hierarchical CuO-ZnO/SiO2 fibrous membranes for water treatment (Adv. Fiber Mater., 2022, 4, 1069); Flexible ceramic fibers for water treatment (Adv. Fiber Mater., 2022, 4, 573; Adv. Fiber Mater., 2022, 4, 129);
7. The reusability and stability of this sample should be measured and evaluated.
Author Response

(The authors gave the same response as above.)

Round 2
Reviewer 3 Report
Authors have sufficiently improved the manuscript by running additional characterization. They also have responded regarding my concerns about isotherm studies. However, I still think that the XRD should be interpreted qualitatively. Please calculate the crystallinity degree:
Refer and cite this article if necessary: https://doi.org/10.1007/s13369-022-06786-6 There you'll find:
"... to determine the change of its crystalline property, we employed a calculation of crystalline index based on the entire diffractogram..."
Author Response
Dear reviewer,
please find our response in the attached pdf file.
Kind regards,
Andre Baldermann

Reviewer 4 Report
The revision is qualified for publication.
Author Response

(The authors gave the same response as above.)
